# Highly Similar Sequences of Mature IgA1 Proteases from *Neisseria meningitidis*, *Neisseria gonorrhoeae* and *Haemophilus influenzae*

**DOI:** 10.3390/pathogens11070734

**Published:** 2022-06-28

**Authors:** David Karlinsky, Yuri Prokopenko, Alexei Zinchenko, Larisa Zhigis, Olga Kotelnikova, Lev Rumsh, Ivan Smirnov

**Affiliations:** Shemyakin-Ovchinnikov Institute of Bioorganic Chemistry, Russian Academy of Sciences, GSP-7, Ulitsa Miklukho-Maklaya, 16/10, 117997 Moscow, Russia; tetrahydrofuran@mail.ru (Y.P.); alezina@mail.ru (A.Z.); zhigis@ibch.ru (L.Z.); ovkot.2003@mail.ru (O.K.); lrumsh@mail.ru (L.R.); ivansmr@inbox.ru (I.S.)

**Keywords:** meningococcal meningitis, vaccine, IgA1 protease, *Neisseria meningitidis*, *Neisseria gonorrhoeae*, *Haemophilus influenzae*, BLAST, sequence alignment

## Abstract

The mature serine-type IgA1 protease from *Neisseria meningitidis* serogroup B strain H44/76 (IgA1pr1_28–1004) is considered here as the basis for creating a candidate vaccine against meningococcal meningitis. In this work, we examine the primary structure similarity of IgA1 proteases from various strains of a number of Gram-negative bacteria (*N. meningitidis*, *Neisseria gonorrhoeae*, *Haemophilus influenzae*) in order to find a structural groundwork for creating a broad-spectrum vaccine based on fragments of this enzyme. BLAST has shown high similarity between the primary structure of IgA1pr1_28–1004 and hypothetical sequences of mature IgA1 proteases from *N. meningitidis* (in 1060 out of 1061 examined strains), *N. gonorrhoeae* (in all 602 examined strains) and *H. influenzae* (in no less than 137 out of 521 examined strains). For these enzymes, common regions of sequence correspond to IgA1pr1_28–1004 fragments 28–84, 146–193, 253–539, 567–628, 639–795 and 811–1004, with identity of at least 85%. We believe that these fragments can be used in the development of a vaccine to prevent diseases caused by pathogenic strains of *N. meningitidis* and *N. gonorrhoeae* as well as a significant number of strains of *H. influenzae*.

## 1. Introduction

Bacterial meningitis, an inflammation of the lining of the brain and spinal cord, is caused by various pathogens. The leading role in the onset of this disease is played by meningococci, pneumococci and *Haemophilus influenzae* [1]. One of the virulence factors of these pathogens is an enzyme they produce, IgA1 protease. Metal-dependent and serine IgA1 proteases are widely known and characterized in detail. Metal-dependent IgA1 proteases are produced by Gram-positive bacteria (*Streptococcus pneumoniae*, *Streptococcus sanguis*, *Streptococcus oralis*, etc.) and serine IgA1 proteases by Gram-negative pathogens (*Neisseria meningitidis*, *Neisseria gonorrhoeae*, *H. influenzae*, etc.) [2,3].

Full-length IgA1 proteases of the serine type are monomeric autotransporters consisting of three functional domains: a signal peptide, a passenger domain and a translocator domain. The N-terminal signal peptide is involved in protein transport across the cytoplasmic membrane. The C-terminal translocator domain transfers the transported domain into the extracellular space. The passenger domain of *N. meningitidis* IgA1 protease is located between the signal peptide and a translocator domain and contains a protease subdomain, γ- and α-peptides [4].

On the outer membrane of the bacterium, IgA1 protease undergoes autocatalytic processing, mainly at the prolyl–serine bonds localized in the γ-peptide [5], which leads to the formation of the mature enzyme sequence. Mature serine chymotrypsin-like IgA1 protease consists of ~1000 amino acid residues, and in the active center it contains the classic catalytic triad characteristic of this family of proteases.

IgA1 proteases have unique substrate specificity, cleaving the prolyl–serine or prolyl–threonine peptide bond within the double octapeptide of the hinge region of human serum and secretory immunoglobulins A1 (IgA1 and S-IgA1) and in several other proteins. One of the functions of bacterial IgA1 proteases is to cleave S-IgA1 on the mucous membranes of the host organism, which leads to a weakening of the protective effect of this protein. Neutralization of IgA1 proteases at this stage of invasion can hinder the adhesion of bacteria to the surface of the mucous membrane and slow down the development of the infectious process. This information, combined with the data on high sequence similarity of IgA1 proteases from *N. meningitidis* (also homologous to IgA1 proteases from at least some strains of *N. gonorrhoeae* and *H. influenzae*) [6], allowed us to consider the IgA1 protease as a promising antigen for the development of an anti-meningococcal vaccine based on it and initiated our research in this direction.

Currently, in public health practice there is a wide range of vaccines against meningococcal meningitis. The effectiveness of these vaccines is generally recognized throughout the world. However, for protection against different serogroups of meningococcus, it is required to use multicomponent vaccines or vaccines against each specific serogroup [7,8,9].

We believe that the development of a single-component vaccine against a wide range of bacterial pathogens with a common virulence factor is still relevant, and the search for appropriate immunologically harmless protective antigens is an important research task [10].

Our team has shown [11,12] that mature IgA1 protease from a live culture of *N. meningitidis* serogroup A, recombinant variants of the enzyme from meningococcus serogroup B strain H44/76 as well as its mutants of the serine residue of the active center protected mice from infection with virulent culture of meningococci of serogroups A, B and C.

In addition, it was found [13,14] that some fragments of the mature enzyme also have immunogenic and protective properties against meningococcal infection. It has also been shown [15] that antibodies formed upon immunization of animals with a mature IgA1 protease are able to bind not only with the secreted protein but also with the surface of meningococcal microbial cells. The results obtained showed that the mature IgA1 protease and some of its fragments can be used as a candidate active ingredient in anti-meningococcal vaccine. This statement is supported by the fact that, on the basis of metal-dependent IgA1 protease from *Streptococcus suis*, a vaccine was created against an infectious disease in pigs caused by the eponymous bacterial pathogen [16].

Nowadays, genome sequence data are available for hundreds of strains of *N. meningitidis*, *N. gonorrhoeae* and *H. influenzae*, so we aimed to investigate highly similar hypothetical sequences of mature IgA1 proteases from these organisms and to find specific highly similar fragments that might be used in the design of protective antigens for a universal vaccine to prevent diseases caused by these bacteria.

## 2. Results and Discussion

### 2.1. Preparing for Calculations

The ability to directly examine IgA1 proteases from known strains of *N. meningitidis*, *N. gonorrhoeae* and *H. influenzae* is hampered by the lack of structured naming of proteins. Therefore, we used the BLAST [17,18] programs to search different strains for sequences that have high similarity with different regions of the full-length IgA1 protease of *N. meningitidis* serogroup B strain H44/76 (IgA1pr1), viewing amino acid and corresponding nucleotide sequences. To be able to obtain data not only on the presence but also on the absence of the desired sequences in certain strains, we selected some lists of basic strains that should be considered for this task. For such lists, we used the “not excluded” strains, for which the proteomes were presented in the Universal Protein Resource [19] in the Proteomes section (see Section 3.1). The table where they are listed indicates whether their genome is represented in the RefSeq database—these statements were not entirely accurate (so we had to conduct separate BLAST for some strains), but they were in general consistent with our choice of databases for BLAST: refseq_genomes for TBLASTN and refseq_protein for BLASTP. In this case, BLASTP was useful for obtaining overview information through reference proteins and TBLASTN for gathering data about individual strains.

### 2.2. Sequence Similarity of the IgA1pr1 Full Chain with Proteins from Different Strains of N. meningitidis

BLASTP of IgA1pr1 in *N. meningitidis* showed a distance tree (see Appendix A) with three types of proteins containing the peptidase_S6 domain. One of these types has high sequence similarity with IgA1pr1 and is named in the reference proteins as “IgA-specific serine endopeptidase autotransporter”. Thus, we assume that in *N. meningitidis*, all proteins highly similar to IgA1pr1 by primary structure are IgA1 proteases. As for the proteins App and MspA [20], also shown in distance tree, their sequences have low similarity with IgA1pr1 (identity of about 34–40% without long regions highly similar to IgA1pr1); therefore, they are not considered further in this article.

TBLASTN of IgA1pr1 in *N. meningitidis* provided data for all 1061 strains belonging to the basic list. In 1060 strains we found nucleotide sequences corresponding to proteins highly similar to IgA1pr1, and only one strain (NM1359) did not show any of these. For 979 strains, the translated protein sequences were similar to IgA1pr1 at Top Identity from 75.59% with full Query cover. Each of the remaining 81 strains had similar data fragmented (with Top Identity from 68.70% to 100.00%—see Table 1); when combined, they had close to the full Query cover.

Here, and in the following results in Section 2.3, we point out when hypothetical protein sequence data is fragmented, but in this study we did not examine the nature of these fragmentations such as actual frameshift mutations or sequencing errors that have resulted in the same exterior (when data is fragmented in one accession), nor do we discuss merging together the results of shotgun sequencing (when data is fragmented in multiple accessions). As a precaution, we still consider fragmented data, but if the IgA1 protease gene in some particular strain is indeed critically damaged, then a positive result is obtained without the vaccine (however, possible non-critically damaged ones can be missed). Similarly, if a found gene is, for some reason, not expressed to produce IgA1 protease, it also counts as a positive result.

It is known from the literature that in *N. meningitidis* IgA1 proteases, the α-peptide region has low homology, and the translocator domain is inside the membrane, which is why we do not consider them as potential components of the vaccine. The signal peptide is cleaved, and thus, for the vaccine, we are interested in the protease domain and, possibly, the γ-peptide (which in various situations remains on the cell surface or is cleaved with the protease domain). Together, the protease domain and the γ-peptide constitute the maximum length mature enzyme [2,4,5]: for IgA1pr1 it is region 28–1004 (IgA1pr1_28–1004). The question of its similarity among strains is of interest for further consideration.

### 2.3. Sequence Similarity of the IgA1pr1 Individual Regions with Proteins from Different Strains of N. meningitidis, N. gonorrhoeae, and H. influenzae

#### 2.3.1. IgA1pr1_28–1004

TBLASTN of IgA1pr1_28–1004 in *N. meningitidis* provided data for all 1061 strains belonging to the basic list, in 1060 of them finding nucleotide sequences corresponding to proteins highly similar to IgA1pr1_28–1004. For 1013 strains, the translated protein sequences were similar to IgA1pr1_28–1004 at Top Identity from 87.20% with full Query cover. Data for these sequences in other 47 strains was fragmented (and for each strain the combination of fragments was close to full Query) with Top Identity from 85.63% to 100.00% (see Table 1). As it was clear from the IgA1pr1 BLAST, in the basic list only strain NM1359 showed no nucleotide sequences corresponding to proteins similar to IgA1pr1_28–1004. This suggests that the vaccine, the active ingredient of which would be the IgA1pr1_28–1004 mutant devoid of proteolytic activity, could provide protection against the vast majority of *N. meningitidis* strains and serogroups. Additionally, such a vaccine could protect against other pathogens in which mature IgA1 protease is more or less homologous to IgA1pr1_28–1004. Here we considered *N. gonorrhoeae* and *H. influenzae*, since they are known to have a large number of strains containing IgA1 proteases that have a highly similar primary structure to IgA1 proteases from *N. meningitidis*.

BLASTP of IgA1pr1_28–1004 in *N. gonorrhoeae* showed a distance tree (see Appendix A) with two branches. One of them contained App proteins (about 34–40% Identity without long regions highly similar to IgA1pr1_28–1004, so we didn’t position them as additional targets). The second branch contained proteins with high Identity, which were likely to be IgA1 proteases, and they were to be viewed more thoroughly. As a result of TBLASTN of IgA1pr1_28–1004 in *N. gonorrhoeae*, out of 602 strains belonging to the basic list, nucleotide sequences corresponding to proteins highly similar to IgA1pr1_28–1004 were found in all strains: in 590 non-fragmented translated sequences were found with Top Identity from 87.10% to 88.51% with full Query cover, in 12—data were fragmented (and for each strain covered full (for 11 strains the combination of fragments was close to full Query, and for one strain the fragments together covered about 89% of the Query) with Top Identity from 76.58% to 97.89% (see Table 1)).

BLASTP of IgA1pr1_28–1004 in *H. influenzae* showed a more complicated picture for the types of found reference proteins (see Appendix A). Hap proteins [21] are clearly shown, with low Identity and without long regions highly similar to IgA1pr1_28–1004. Among the proteins with high Identity some are shown to be IgA-B2 proteases, and among proteins with Identity of roughly 51–59% some are shown to be IgA-A1, IgaA-A2 and IgA-B1 proteases [22]. TBLASTN of IgA1pr1_28–1004, considering the refseq_genomes database, showed data for 521 out of 536 strains belonging to the basic list. High sequence similarity with IgA1pr1_28–1004 was shown for translated nucleotide sequences from 137 strains. Among these, in 124 strains, translated sequence data was non-fragmented, their sequence similarity with IgA1pr1_28–1004 described as Top Identity from 85.21% to 93.58% with a full Query cover, and in 13 strains the sequence data was fragmented (and for each strain the combination of fragments was close to full query) with Top Identity from 79.33% to 93.54% (see Table 1). The remaining 384 strains, with this approach, showed no data for sequences highly similar to IgA1pr1_28–1004. We did not perform an additional search for individual strains of *H. influenzae*, since a detailed consideration of *H. influenzae* strains and serotypes against which a potential vaccine would be effective is beyond the scope of the current study.

Thus, we can assume that a vaccine based on IgA1pr1_28–1004 will protect against *N. meningitidis*, *N. gonorrhoeae* and, partially, *H. influenzae*. However, mature IgA1 proteases have a high molecular mass and are highly immunogenic. To reduce the immune load on an organism, it is desirable to use shorter proteins; therefore, we studied in more detail the similarity of individual regions of the amino acid sequence of IgA1 proteases from various sources.

#### 2.3.2. Short Fragments

Alignment of the results of TBLASTN of IgA1pr1 fragment 1–1004 (mature enzyme with a signal peptide), jointly in *N. meningitidis*, *N. gonorrhoeae* and *H. influenzae*, showed that, within IgA1pr1_28–1004, regions 28–84, 146–193, 253–539 and 567–1004 have an increased sequence similarity with regions of IgA1 proteases from different strains of these organisms (see Appendix A).

BLASTP showed that low complexity regions inside the area 28–1004 of IgA1pr1 are situated at sections 552–562, 629–638 and 796–810, and within the alignment with this area, reference proteins with high sequence similarity from *N. meningitidis*, *N. gonorrhoeae* and *H. influenzae* have low complexity regions only inside the corresponding aligned sections.

Therefore, for the further study we propose fragments 28–84, 146–193, 253–539 and (since fragment 567–1004 is divided by low complexity regions) fragments 567–628, 639–795 and 811–1004 of IgA1pr1_28–1004. TBLASTN results for these fragments are presented in Table 2. It includes the results for those strains in which the obtained sequence data are not fragmented at the region corresponding to IgA1pr1_28–1004. It can be seen from this table that, in all the strains considered, there are fragments that do not contain low complexity regions and have a Top Identity of at least 85% with full Query cover, corresponding to regions 28–84, 146–193, 253–539, 567–628, 639–795 and 811–1004 of IgA1pr1_28–1004.

Figure 1 shows mapping of these proposed highly similar regions (see Table 2) of tested IgA1 protease sequences to an AlphaFold2 [23,24] model of IgA1 protease from *N. gonorrhoeae* strain ATCC 700825/FA 1090 (https://alphafold.ebi.ac.uk/entry/Q5F9W2, accessed at 9 June 2022), region 28–1018. This protein sequence is shown aligned in Multiple Sequence Alignment in Appendix A where the strain is named as FA 1090, and these regions are: 28–84, 147–194, 264–549, 577–638, 649–805 and 821–1018. In Figure 1, these conservative regions are shown in different colors and one can see that they are presented on the protein surface. They can be considered for further investigation (in silico, in vitro, in vivo) when designing an active ingredient of a universal vaccine against *N. meningitidis*, *N. gonorrhoeae* and some strains of *H. influenzae*.

## 3. Materials and Methods

### 3.1. Basic Lists of Strains

The strains presented on the Universal Protein Resource [19] (https://www.uniprot.org/, accessed 30 March–3 April 2020) in the Proteomes section were considered as those of interest to us—namely, the lists of strains for which proteomes were specified as “not-excluded”: 1061 strains of *N. meningitidis*, 602 strains of *N. gonorrhoeae* and 536 strains of *H. influenzae*.

### 3.2. Nucleotide and Amino Acid Sequences for Input Data

The nucleotide sequences of the coding gene of the full-length IgA1 protease of *N. meningitidis* serogroup B strain H44/76 (IgA1pr1) are presented at the National Center for Biotechnology Information [25] (https://www.ncbi.nlm.nih.gov/ (accessed on 28 May 2020)) website under the codes NCBI Reference Sequence: NC_017516.1 from 1574334 to 1579040 and NCBI Reference Sequence: NZ_AEQZ01000045.1 from 47053 to 42347. Their corresponding amino acid sequence is presented as NCBI Reference Sequence: WP_002221261.1.

In this research, we used the RefSeq databases [26]: refseq_protein and refseq_genomes. Additionally, nucleotide sequences from genomes of a number (see Appendix B) of *N. meningitidis* and *N. gonorrhoeae* strains were obtained from the European Nucleotide Archive [27] (https://www.ebi.ac.uk/ena/browser/home (accessed on 10 June 2020)).

### 3.3. BLAST

In this work, we used two programs from the BLAST [17,18] family at the National Center for Biotechnology Information website: BLASTP and TBLASTN (accessed through 2020, BLASTP distance trees are rebuilt in Appendix A at 6 June 2022). The “Max target sequences” parameter was always set to the maximum available (5000 or 20,000). All parameters that are not mentioned specifically were used with the default values. The alignment results were viewed in the “Alignment view: Flat query-anchored with dots for identities” option with the Line length parameter equal to 60 or 150.

In total, within the framework of this study, the following was carried out.

BLASTP (filtering low complexity regions), considering refseq_protein database:IgA1pr1 in *N. meningitidis*;IgA1pr1_28–1004 separately in *N. gonorrhoeae* and *H. influenzae*.

TBLASTN (not filtering low complexity regions):IgA1pr1 in *N. meningitidis*, considering the refseq_genomes database and genomes of individual strains;IgA1pr1_28–1004 (a) in *N. meningitidis*, considering the refseq_genomes database and genomes of individual strains; (b) in *N. gonorrhoeae*, considering the refseq_genomes database and genomes of individual strains; (c) in *H. influenzae*, considering the refseq_genomes database;IgA1pr1 fragment 1–1004 jointly in *N. meningitidis*, *N. gonorrhoeae* and *H. influenzae*, considering the refseq_genomes database;IgA1pr1 fragments 28–84, 85–145, 146–193, 194–252, 253–539, 540–566, 567–1004, 567–628, 639–795, 811–1004 separately in *N. meningitidis*, *N. gonorrhoeae* and *H. influenzae* considering the refseq_genomes database.

## 4. Conclusions

In this study, we have shown that the full-length IgA1 protease hypothetically (taking into consideration that some data is fragmented) has high sequence similarity in 1060 of the 1061 examined strains of *N. meningitidis*. This statement is also true for the mature protein region, which has high sequence similarity with the corresponding regions of hypothetical IgA1 proteases from all 602 examined strains of *N. gonorrhoeae* and 137 of 521 examined strains of *H. influenzae*. In the mature protein, highly similar fragments of the primary structure are located in the intervals 28–84, 146–193, 253–539, 567–628, 639–795 and 811–1004 (according to the numbering of the IgA1 protease from *N. meningitidis* strain H44/76), and these are what we suggest for further consideration in designing an active ingredient for the vaccine against *N. meningitidis* and *N. gonorrhoeae* as well as a range of strains of *H. influenzae*. In perspective, the next steps are: matching the sequence similarity of potential candidates with proteins from other organisms, modeling tertiary structures and searching for epitopes.

## Figures and Tables

**Figure 1 pathogens-11-00734-f001:**
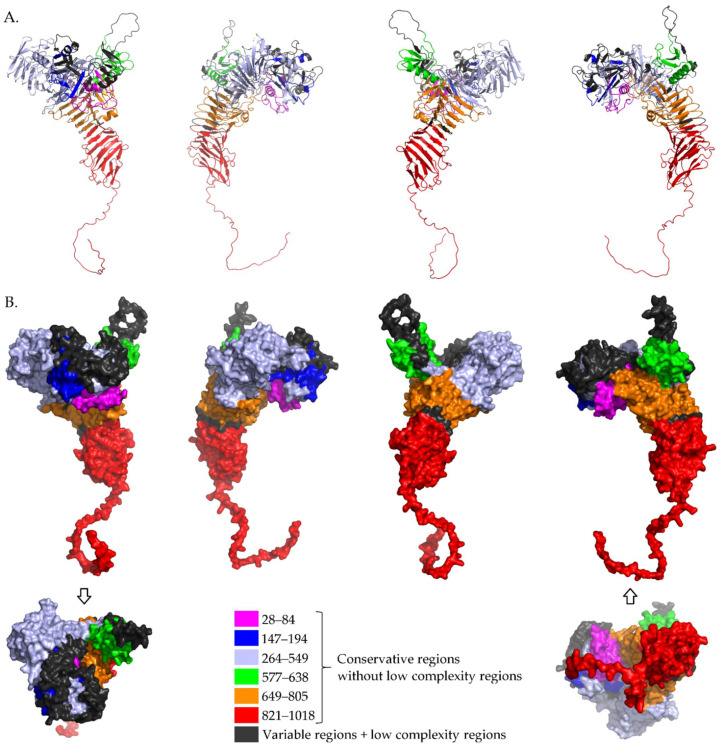
Mapping conservative regions (without low complexity regions) of tested IgA1 proteases to an AlphaFold2 model of IgA1 protease from *N. gonorrhoeae* strain ATCC 700825/FA 1090, region 28–1018. (**A**) Cartoon view and (**B**) surface view.

**Table 1 pathogens-11-00734-t001:** TBLASTN results for a full-length (IgA1pr1) and a max-length mature (IgA1pr1_28–1004) IgA1 proteases from *Neisseria meningitidis* serogroup B strain H44/76: IgA1pr1 in *N. meningitidis* and IgA1pr1_28–1004 in *N. meningitidis*, *Neisseria gonorrhoeae* and *Haemophilus influenzae*.

Data Fragmentation Level	IgA1pr1 in *N. meningitidis*		IgA1pr1_28–1004 in *N. meningitidis*		IgA1pr1_28–1004 in *N. gonorrhoeae*		IgA1pr1_28–1004 in *H. influenzae*	
	Number of Strains (from 1061)	Top Identity	Number of Strains (from 1061)	Top Identity	Number of Strains (from 602)	Top Identity	Number of Strains (from 521)	Top Identity
Non-fragmented	979	75.59–100.00%	1013	87.20–100.00%	590	87.10–88.51%	124	85.21–93.58%
Fragmented in one accession	68	68.70–100.00% *	39	85.63–100.00% *	11	83.97–97.89% *	12	79.33–93.54% *
Fragmented in multiple accessions	13	68.83–100.00% *	8	88.74–100.00% *	1	76.58–94.58% *	1	91.08% *

* For fragmented data, the Top Identity values are local and, due to the possible inclusion of redundant areas in the comparison, are less accurate.

**Table 2 pathogens-11-00734-t002:** TBLASTN results for IgA1pr1_28–1004 fragments.

**A.** Top Identity and corresponding Query cover of sequences in *N. meningitidis*.										
Sequence range	28–84	85–145	146–193	194–252	253–539	540–566	567–1004	567–628	639–795	811–1004
Number of strains (from 1887)	1887	1887	1887	1887	1887	893	1887	1887	1887	1887
MIN Top Identity	98.25%	57.38%	87.50%	56.86%	88.85%	96.30%	92.76%	88.71%	95.54%	88.89%
AVG Top Identity	99.99%	87.14%	96.76%	86.60%	96.01%	99.90%	97.52%	95.63%	98.64%	97.23%
MAX Top Identity	100.00%	100.00%	100.00%	100.00%	100.00%	100.00%	100.00%	100.00%	100.00%	100.00%
MIN Query cover of Top Identity	100%	98%	100%	86%	100%	100%	100%	100%	100%	100%
AVG Query cover of Top Identity	100%	99%	100%	96%	100%	100%	100%	100%	100%	100%
MAX Query cover of Top Identity	100%	100%	100%	100%	100%	100%	100%	100%	100%	100%
**B.** Top Identity and corresponding Query cover of sequences in *N. gonorrhoeae*.										
Sequence range	28–84	85–145	146–193	194–252	253–539	540–566	567–1004	567–628	639–795	811–1004
Number of strains (from 683)	683	683	683	683	683	683	683	683	683	683
MIN Top Identity	96.49%	59.02%	85.42%	54.90%	90.24%	92.59%	92.99%	93.55%	94.90%	89.39%
AVG Top Identity	98.93%	59.08%	87.53%	56.92%	92.43%	99.39%	94.07%	96.81%	97.11%	90.20%
MAX Top Identity	100.00%	100.00%	100.00%	100.00%	96.86%	100.00%	94.57%	98.39%	97.45%	91.41%
MIN Query cover of Top Identity	100%	98%	100%	86%	100%	100%	100%	100%	100%	100%
AVG Query cover of Top Identity	100%	98%	100%	86%	100%	100%	100%	100%	100%	100%
MAX Query cover of Top Identity	100%	100%	100%	100%	100%	100%	100%	100%	100%	100%
**C.** Top Identity and corresponding Query cover of sequences in *H. influenzae*.										
Sequence range	28–84	85–145	146–193	194–252	253–539	540–566	567–1004	567–628	639–795	811–1004
Number of strains (from 154)	154	154	154	154	154	0	154	154	154	154
MIN Top Identity	100.00%	59.02%	89.58%	56.86%	88.50%	-	92.31%	88.71%	95.54%	90.91%
AVG Top Identity	100.00%	71.28%	92.83%	67.67%	91.20%	-	93.68%	90.91%	96.00%	92.53%
MAX Top Identity	100.00%	98.36%	100.00%	91.53%	94.43%	-	95.48%	100.00%	96.82%	93.43%
MIN Query cover of Top Identity	100%	98%	100%	86%	100%	-	100%	100%	100%	100%
AVG Query cover of Top Identity	100%	99%	100%	90%	100%	-	100%	100%	100%	100%
MAX Query cover of Top Identity	100%	100%	100%	100%	100%	-	100%	100%	100%	100%

Green color indicates areas that, for all the strains considered in this paragraph, have Top Identity of at least 85% with full Query cover and do not contain low complexity regions.

## Data Availability

Main data for this study is available in the manuscript and online Appendix A. Additional data is available upon a reasonable request to D.K.

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
