# Peer review of "Highly Similar Sequences of Mature IgA1 Proteases from Neisseria meningitidis, Neisseria gonorrhoeae and Haemophilus influenzae"

_pathogens, 2022, doi:10.3390/pathogens11070734_

Round 1

Reviewer 1 Report

This was an interesting paper looking at the similarity of the IgA proteases between three pathogens with the intention of investigating peptides as vaccine candidates. The analysis was thorough but not very in depth. There are two aspects of the paper that could improve its impact: there were allusions to other groups of similar proteins like Msp, App etc. I would have liked to see these phylogenies in the paper as this is a nicer way of showing distance relationships within large datasets. Also, having identified regions of high homology, I think their relevance to a vaccine study would have been best displayed by mapping the peptides onto a molecular model of the enzyme. In this way, it could be reasonably discussed whether the conserved regions were on the outside of the protein or inside. Many peptides could be on the inside and be of no value at all for a vaccine approach. Selecting the peptides using this approach would add more value to the current study.

Author Response

Thank You very much for the review!

Point 1: There are two aspects of the paper that could improve its impact: there were allusions to other groups of similar proteins like Msp, App etc. I would have liked to see these phylogenies in the paper as this is a nicer way of showing distance relationships within large datasets.

Response 1: In accordance with Your comments, we have added three phylogenetic trees to the Supplement (Supplementary Figures S1, S2, S3 - please see the attachment). The BLASTP results demonstrate that proteins like Hap, MspA, App etc. have a rather low sequence similarity with examined IgA1 proteases, thus we exclude these proteins from our consideration. Additionally, we would like to note that sequence data for these proteins should be used after additional validation of the names, as we did for the considered IgA1 proteases when we performed tblastn calculations.

Point 2: Also, having identified regions of high homology, I think their relevance to a vaccine study would have been best displayed by mapping the peptides onto a molecular model of the enzyme. In this way, it could be reasonably discussed whether the conserved regions were on the outside of the protein or inside. Many peptides could be on the inside and be of no value at all for a vaccine approach. Selecting the peptides using this approach would add more value to the current study.

Response 2: We thank You for this comment, and we add Figure S4 to the Supplement (please see the attachment). We used the AlphaFold2 model of an IgA1 protease from Neisseria gonorrhoeae strain ATCC 700825 / FA 1090 (https://alphafold.ebi.ac.uk/entry/Q5F9W2) and mapped the high-homologous regions onto this structure. The sequence of selected IgA1 protease is aligned with other IgA1 proteases from N. meningitidis, N. gonorrhoeae, H. influenzae (see IgA1 protease from N. gonorrhoeae strain FA 1090 in Supplementary Table S1 - please see the attachment). We have shown that the major part of the enzyme surface belongs to high-homologous regions and thus can be considered as regions for potential vaccines. We have added an appropriate text to the manuscript (please see the attachment, changes that are marked yellow in the manusctipt file).

Attachment: https://cloud.mail.ru/public/pJ1d/LXRcQkJH1 , password: neisseria .

With best regards,

David Karlinsky

Reviewer 2 Report

This study is an extension of previous work by this team, which has demonstrated that the N. meningitidis IgA1 protease is a potential vaccine candidate. This manuscript describes work that is entirely database and bioinformatics driven to identify that the N. meningitidis IgA1 protease is highly conserved between strains of Nm, as well conserved with N. gonorrhoeae (a closely related organism) and a subset of Haemophilus influenzae strains. The rationale for focusing on the protease domain and possibly the gamma-peptide, which are expected to be on the cell surface or cleaved, is clearly explained. Further investigation identified selected fragments which could be used as the basis of a vaccine with the potential to cover Nm, Ng and a portion of Hi strains. The work is clearly described and follows a logical path.

Minor comments;

Line 63: please remove “which significantly increases the antigen load on the human body”. Many vaccines are complex – even comprising of whole live attenuated organisms (e.g. polio) and the antigen load administered via vaccination – or even multiple inoculations of the same vaccine – is relatively trivial compared to every day microbial exposure.

Line 67: Note that there is also evidence that single component vaccines are less effective and can become ineffective over time due to antigen drift and declining antibody titres – e.g. the acellular pertussis vaccine. It would perhaps be more accurate to indicate that cruder vaccine formulations tend to be more reactogenic, therefore there is a need to produce better defined/refined vaccines.

Line 283: not sure what is meant by “This enzyme is perspective for designing….”?

Author Response

Thank You very much for the review!

In accordance with Your comments, the following changes have been made:

Point 1:

Line 63: please remove “which significantly increases the antigen load on the human body”. Many vaccines are complex – even comprising of whole live attenuated organisms (e.g. polio) and the antigen load administered via vaccination – or even multiple inoculations of the same vaccine – is relatively trivial compared to every day microbial exposure.

Line 67: Note that there is also evidence that single component vaccines are less effective and can become ineffective over time due to antigen drift and declining antibody titres – e.g. the acellular pertussis vaccine. It would perhaps be more accurate to indicate that cruder vaccine formulations tend to be more reactogenic, therefore there is a need to produce better defined/refined vaccines.

Response 1: 

Manuscript text changed:

Line 60:

"Currently, in public health practice there is a wide range of vaccines against menin-gococcal meningitis. The effectiveness of these vaccines is generally recognized through-out the world. However, for protection against different serogroups of meningococcus, it is required to use multicomponent vaccines, or vaccines against each specific serogroup [7–9].

We believe that the development of a single-component vaccine against a wide range of bacterial pathogens with a common virulence factor is still relevant, and the search for appropriate immunologically harmless protective antigens is an important research task [10]."

Point 2: Line 283: not sure what is meant by “This enzyme is perspective for designing….”?

Response 2: Manuscript text changed. In the Conclusions section, phrase

“This enzyme is perspective for designing an active ingredient of the universal vaccine against N. meningitidis and N. gonorrhoeae, as well as a range of strains of H. influenzae.”

is removed. Instead,

“… against N. meningitidis and N. gonorrhoeae, as well as a range of strains of H. influenzae”

is added to the next phrase.

With best regards,

David Karlinsky

Round 2

Reviewer 1 Report

Thank you for responding to my comments. However, the supplementary files Supplementary Fig 2 and 3 have not helped me very much. I am not sure why but the phylogenetic trees I see are not labelled at all or coloured in anyway - so I do not understand what is in them. This may well be an error in their formatting- so this needs to be fixed. It may well need to be converted to a .png file.

The 3D model was good but I did not understand what the colouring was about. What I was looking for was the mapping of your peptides onto the model to show that they were surface exposed. As far as I can see from the legend- this is not what the colouring was about? In other words, I expected to see  fragments 28-84, 146-193, 253-539, 567-628, 639- 20 795 and 811-1004 mapped onto the model in different colours.  The reason for this is that many neisserial surface exposed proteins are known to have conserved and non-conserved regions, but typically the conserved regions are buried. So my concern is that your peptides may not be useful in vaccine design if they are conserved but not surface exposed.

Author Response

Thank You very much for Your comments.

Point 1: Thank you for responding to my comments. However, the supplementary files Supplementary Fig 2 and 3 have not helped me very much. I am not sure why but the phylogenetic trees I see are not labelled at all or coloured in anyway - so I do not understand what is in them. This may well be an error in their formatting- so this needs to be fixed. It may well need to be converted to a .png file.

Response 1: It seems that pdf files can be viewed differently in different programs. I thought that this file format is free from such surprises. We are sending the png files instead.

Point 2: The 3D model was good but I did not understand what the colouring was about. What I was looking for was the mapping of your peptides onto the model to show that they were surface exposed. As far as I can see from the legend- this is not what the colouring was about? In other words, I expected to see  fragments 28-84, 146-193, 253-539, 567-628, 639- 20 795 and 811-1004 mapped onto the model in different colours.  The reason for this is that many neisserial surface exposed proteins are known to have conserved and non-conserved regions, but typically the conserved regions are buried. So my concern is that your peptides may not be useful in vaccine design if they are conserved but not surface exposed.

Response 2: Oh, I hope that now I get it. About the model itself, however, we still presume that it is better to use AlphaFold Protein Structure Database model of IgA1 protease from N. gonorrhoeae strain ATCC 700825 / FA 1090. This protein is mentioned in the Supplementary Table S1, with the discussed regions having high homology, so using it should be totally fine. If we introduce an IgA1pr1_28-1004 tertiary structure model, we will have to discuss its quality and bring appropriate pictures, which might be a bit too far outside of the paper scope. In the manuscript, Figure 1 now shows these mapped peptides.

With best regards,

David Karlinsky